# IMARA: A mother-daughter group randomized controlled trial to reduce sexually transmitted infections in Black/African-American adolescents

Geri R. Donenberg[1,2,3]*, Ashley D. Kendall[1,2], Erin Emerson[1,2,3], Faith E. Fletcher[4], Bethany C. Bray[1], Kelly McCabe[1,2]

1 Center for Dissemination and Implementation Science, Department of Medicine, College of Medicine, University of Illinois at Chicago, Chicago, Illinois, United States of America, 2 Healthy Youths Program, Department of Medicine, College of Medicine, University of Illinois at Chicago, Chicago, Illinois, United States of America, 3 Community Outreach Intervention Projects, School of Public Health, Chicago, Illinois, United States of America, 4 School of Public Health, University of Illinois at Chicago, Chicago, Illinois, United States of America

* gerid@uic.edu

**Data Availability Statement:** All relevant data are within the manuscript and its Supporting Information files.

## Abstract

Black/African-American girls are infected with sexually transmitted infections (STIs) at higher rates than their White counterparts. This study tested the efficacy of IMARA, a mother-daughter psychosocial STI/HIV prevention program, on adolescent Black/African-American girls' incident STIs at 12 months in a 2-arm group randomized controlled trial. Black/African-American girls 14–18 years old and their primary female caregiver were eligible for the study. Girls provided urine samples to test for *N. gonorrhoeae*, *C. trachomatis*, and *T. vaginalis* infection at baseline and 12-months. Mother-daughter dyads were randomly assigned to IMARA (n = 118) or a time-matched health promotion control program (n = 81). Retention at 12-months was 86% with no difference across arms. Both interventions were delivered over two consecutive Saturdays totaling 12 hours. Girls who received IMARA were 43% less likely to contract a new STI in the 12-month post-intervention period compared with those in the health promotion control program (*p* = .011). A secondary follow-up intent-to-treat analysis provided additional support for the protective effect of IMARA, albeit with a similar magnitude of 37% (*p* = .014). Findings provide early evidence for IMARA's efficacy, such that IMARA protected against STIs at 12-months among adolescent Black/African-American girls. Future research should examine the mechanisms associated with reduced STIs.

## Introduction

Youth ages 15–24 years-old account for 50% of new sexually transmitted infections (STIs) annually [1, 2], yet rates vary by race/ethnicity. Black/African-American adolescent females

**Funding:** This research was supported by a grant (R01MD006198) from the National Institute of Minority Health and Health Disparities (https://www.nimhd.nih.gov) to GD. The funders had no role in study design, data collection and analysis, decision to publish, or preparation of the manuscript.

**Competing interests:** The authors have declared that no competing interests exist.

are infected with Chlamydia, gonorrhea, and HIV at 4.5, 10.3, and 20 times that of their White counterparts, respectively [3, 4]. Untreated STIs can lead to serious reproductive health consequences, including pelvic inflammatory disease and vulnerability to HIV acquisition, and high medical costs [5]. Innovative approaches to alleviate STIs among Black/African-American girls are a public health imperative.

Existing STI/HIV prevention programs for Black/African-American adolescent girls and women report small effect sizes, short-term effects, and non-behavioral outcomes (e.g., attitudes, knowledge) [6–9]. Most focus on individual-level behavior change, but social factors influence sexual behavior and STI acquisition [10]. Female caregivers (hereafter referred to as "mothers") influence daughters' risky sexual behavior, suggesting the mother-daughter relationship may be leveraged to reduce daughters' STI risk. Maternal warmth, support and attachment, as well as positive (e.g., open, receptive) mother-daughter communication, are each associated with girls' reduced sexual risk [11–14], including among Black/African-American adolescent girls [11]. A recent study showed strong mother-daughter relationships and open communication predicted less sexual risk taking among Black/African-American adolescent girls two years later [11]. In light of these findings, mothers may be effective collaborators in preventing STI in daughters [6, 15, 16].

Informed, Motivated, Aware, and Responsible about AIDS (IMARA) is a mother-daughter STI/HIV prevention program for Black/African-American girls guided by a social-personal framework [17] and derived from three evidence-based interventions: Sisters Informing Sisters about Topics on AIDS (SISTA) [18], Sistering, Informing, Healing, Living, and Empowering (SiHLE) [19], and Strengthening the Youth Life Experience (Project STYLE) [20]. SISTA and SiHLE address cultural influences on Black/African-American adult women's (SISTA) and adolescent girls' (SiHLE) sexual behavior, labor and power gender divisions, and gender-specific standards for "appropriate" sexual conduct in heterosexual relationships. Randomized controlled trials (RCTs) of SISTA and SiHLE revealed efficacy and areas on which to further strengthen outcome effects. For example, SISTA demonstrated improved self-reported condom use, partner communication, and assertiveness in adult Black/African-American women [18], and SiHLE showed improved condom use and fewer sex partners in adolescent Black/African-American girls [19]. However, neither SISTA nor SiHLE address mental health or the mother-daughter relationship and communication, and findings from SISTA do not report biological outcomes. Project STYLE is complementary to SISTA and SiHLE; it addresses STIs while emphasizing the role of mental health (having been designed for adolescents in psychiatric care) and demonstrated improved parent-teen sexual communication, parental monitoring, and adolescent reported condom use [20]. However, Project STYLE does not focus on the role of gender or racial/ethnic empowerment or report biological outcomes. IMARA was therefore designed to address the gaps in each program by combining their strengths to reduce STIs in Black/African-American adolescent girls.

This study sought to provide the first test of IMARA on Black/African-American adolescent girls' STI acquisition over one-year. As detailed below, 199 mother-daughter pairs were randomly assigned to IMARA or a time-matched health promotion control program. We hypothesized that girls who received IMARA would be less likely to contract a new STI at 12-months than girls in the control group. The study is positioned to provide early but rigorous evidence for an innovative approach to alleviating STIs among Black/African-American adolescent girls, and lay the foundation for future investigation into the mechanistic pathways underlying STI prevention in this group.

## Methods

### Procedures

Participants were 199 Black/African-American girls 14–18 years-old (M = 16.02; SD = 1.35) and their female caregivers. Girls and their caregivers were recruited through two strategies to maximize generalizability and represent a broad range of mental health problems from 11/2012–04/2016. The final follow-up assessment was completed on 06/2017. First, dyads were recruited from mental health clinics, whereby clinic staff hired by the study served as liaisons to the research team. The clinic liaisons identified and presented the study to eligible mothers and/or daughters, and with familial consent, provided their contact information to the research team. This approach was used to enroll 41% of the sample. Second, research staff conducted street outreach, distributing study fliers at community programs, street fairs, and school fairs, collecting names and numbers of interested families, and successfully enrolled 59% of the sample. Inclusion/exclusion criteria was the same across recruitment venues/approaches. In both cases, staff followed up by telephone to provide more information and schedule baseline assessments and the intervention. Upon arrival at the research site, mothers and daughters met separately with researchers to review consent and assent forms, including information about trial design (e.g., random assignment) and confidentiality of girls' STI results (i.e., mothers learned they would not be told). The baseline assessment was completed within 8 weeks pre-intervention to allow time to assemble groups. However, 54% of dyads completed the baseline the same morning of day 1 of the IMARA/health promotion program.

On the morning of workshop day 1, dyads were randomly assigned to the IMARA (n = 118) or the health promotion control program (n = 81) as follows. Upon arrival, daughters selected a slip of paper from a bag that indicated their group assignment (IMARA vs. health promotion control). Dyads were not assigned to groups in advance of workshop day 1 to prevent attendance bias due to group assignment. We employed two randomization strategies depending on the number of dyads present on workshop day 1. Where $\geq 8$ dyads were present at the start of the workshop (52% of dyads), we used a 1:1 randomization ratio; girls selected the slips of paper and were evenly randomized to treatment arms. Where $\leq 7$ dyads were present (48%), we delivered a single program to ensure a sufficient group size. There was one exception where 5 dyads were randomized to two arms. The project director, but not the recruiters, were aware which intervention would be delivered in the event there were too few dyads ($< 8$) to randomize to both arms. In general, we alternated between IMARA and the health promotion condition, but in the final year of recruitment we favored IMARA to enhance analytic power to detect treatment effects and to provide a large enough sample to evaluate mechanisms associated with key outcomes in IMARA. Although originally unplanned, these strategies led to an overall randomization ratio of approximately 1.5:1 favoring IMARA. The CONSORT diagram (Fig 1) illustrates the number of dyads randomized to arms and their retention at 12-month follow-up.

Both programs were delivered in groups ranging from 1–9 dyads over two consecutive Saturdays at the research site for a total of 12 hours (approximately 6 hours each day). Dyads completed a 12-month assessment post-baseline either at the research site or a location convenient for the family. Compensation for all research activities totaled $165 per participant plus reimbursement for travel. The study was approved (#2011–0263) by the University of Illinois at Chicago on 06/08/2011. This trial is registered at Clinicaltrials.gov (NCT02958813). Registration was delayed until 1½ months following the publication of the 2016 Final Policy on the Dissemination of NIH-funded Clinical Trial Information. Up to that point, NIH-funded studies were not required to register, and it was unclear whether the current behavioral study qualified because it did not involve experimental drugs. The Final Rule's publication expanded and

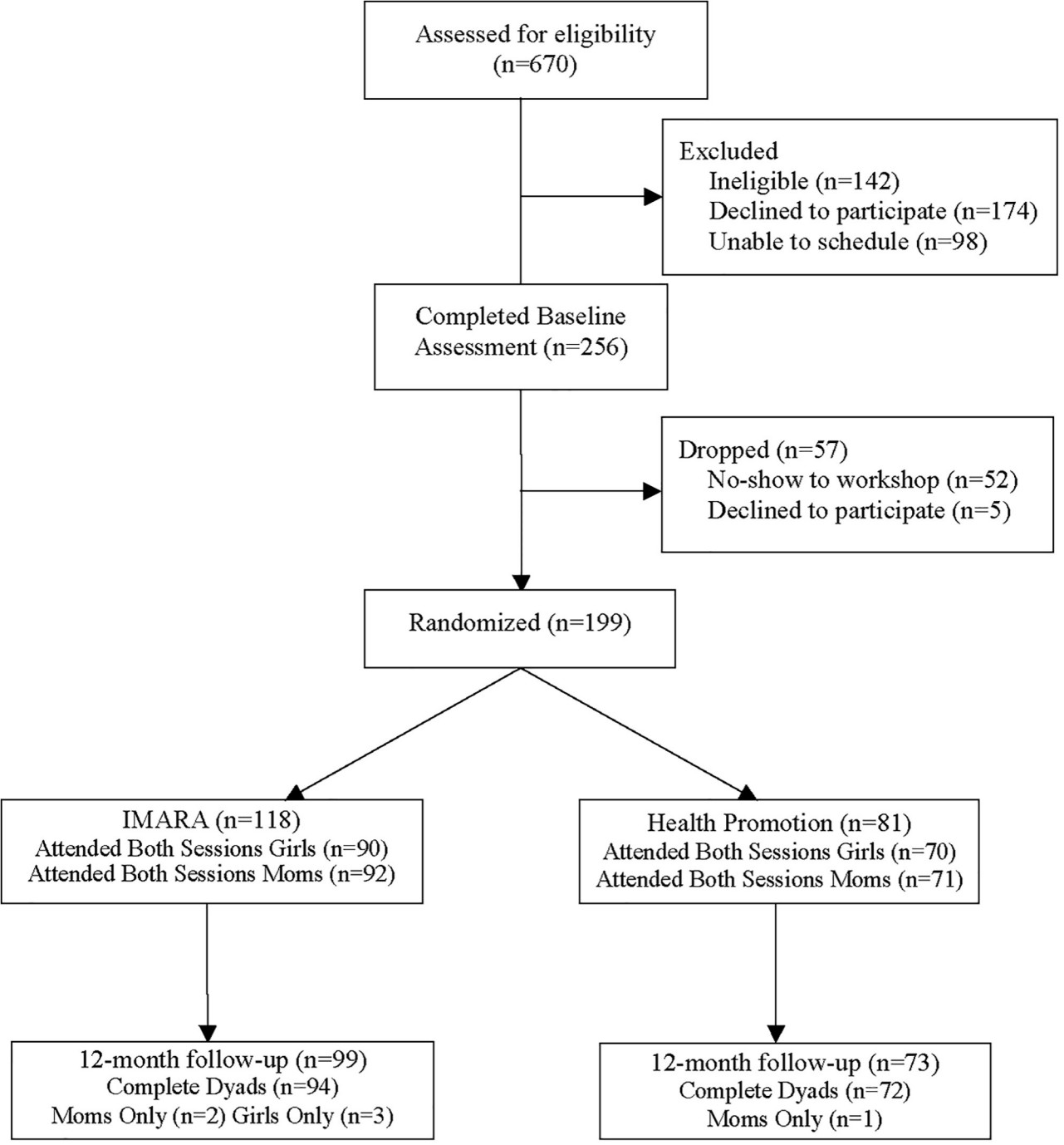

**Fig 1. Consolidated standards of reporting trials (CONSORT) summary of participant enrollment and retention at 12-month follow-up.**

clarified registration requirements, thereby prompting registration. The authors confirm that all ongoing and related trials for this intervention are registered at Clinicaltrials.gov.

## Participants

Adolescent girls were eligible if they: a) were 14–18 years old; b) self-identified as Black/African-American; c) had a primary female caregiver ("mother") who lived with them;

d) understood and provided consent/assent; e) spoke English (measures were normed for English speakers and the intervention was delivered in English); and f) had parental consent. Mothers were eligible if they: a) self-identified as Black/African-American; b) identified as the adolescent's primary female caregiver; c) were > 18 years old; d) understood and provided consent; and e) spoke English (measures were normed for English speakers and the intervention was delivered in English). Girls were excluded if they were actively psychotic or severely mentally ill. No girls were excluded based on this criteria.

## Measures

The AIDS Risk Behavior Assessment (ARBA) [21] was administered at baseline with other measures not included in this study. The ARBA is a widely-used computerized voice-assisted self-administered interview of sexual behavior. Girls reported whether they ever had vaginal/anal sex (yes/no). Those who indicated lifetime vaginal/anal sex were prompted to indicate if they had vaginal/anal sex in the past six months (yes/no), and if they used a condom at last vaginal/anal sex (yes/no).

At baseline and 12-months, girls provided urine samples to screen for *N. gonorrhoeae*, *C. trachomatis*, and *T. vaginalis* via nucleic acid amplification testing. We used the Abbott Real Time CT/NG assay, and the Taq-Man PCR for TV. We did not test for HIV given the low prevalence in adolescents, instead using STIs as a proxy for unprotected sex. STI-positive girls were offered free, single-dose, directly-observed therapy. Consistent with Illinois law, expedited partner therapy was offered. The study physician was unable to confirm partner therapy, but treatment was confirmed for 92% (n = 33/36) of girls at baseline: 31% were treated by the study physician, and 61% were treated by another source verified by research staff via conversations with providers, name of the provider or clinic, and/or description of the medication. Treatment rates did not differ by intervention arm. We were unable to verify treatment of three participants (two assigned to IMARA and one to the health promotion program), and thus their data were excluded from the outcome analyses that relied on known STI status.

## Intervention and health promotion control programs

IMARA and the health promotion control programs were similarly structured. Both programs occurred in groups over two workshop days (12 hours total) separated by one week. Parts of the curriculum were delivered separately to mothers and daughters covering parallel content, and parts of the curriculum brought dyads together in a single group. Participants in the two programs did not overlap. For each program, two facilitators co-led the mothers' group and two co-led the girls' group. All four facilitators co-led mother-daughter joint activities.

**Informed motivated aware and responsible about AIDS.** IMARA was derived from three evidence-based interventions (SISTA; SiHLE; Project STYLE), guidance from a long-standing community advisory board, and pilot testing. Activities from each of the three curricula were evaluated and selected for inclusion based on the goals of the intervention. For example, the parent-teen communication components from Project STYLE were included to strengthen mother-daughter communication, whereas the female empowerment content of SISTA and SiHLE was incorporated to address gender and ethnic pride [18, 19]. Pilot testing with 22 caregiver-daughter dyads provided strong evidence of acceptability and feasibility by girls and their caregivers.

IMARA's curriculum focuses on strengthening mother-daughter relationships and communication, particularly regarding STI/HIV prevention and safer sexual behavior, increasing self-efficacy to use condoms, improving maternal monitoring, promoting pride in Black/African-American culture and encouraging gender empowerment. Mothers and daughters learn

and practice assertive communication through role-plays and games, discuss healthy versus unhealthy relationships with peers and partners, receive facts about STIs and HIV, consider the role of media in Black/African-American girls' self-image, and identify the impact of mental health on risky sexual behavior.

Morning and afternoon sessions begin with all mothers and daughters together in the same group and an icebreaker to enhance ethnic and gender pride. Next, mothers and daughters are separated into their own groups and discuss the impact of marketing and music on perceptions of Black/African-American adolescent girls. Mothers develop monitoring plans to augment oversight of their daughters, and girls identify triggers (people, places, moods, situations) of risk behavior and create personalized plans to manage the triggers. In separate groups, mothers and daughters review safe sex knowledge and attitudes and learn new skills (e.g., condom use). Interactive and experiential activities are designed to catalyze discussions of physical, emotional, and sexual abuse in relationships and the impact of emotions and relationship power imbalances on STI/HIV-risk. Mothers and daughters come together for joint activities designed to strengthen mothers' credibility as a resource for STI/HIV prevention and facilitate communication skills practice. Dyads discuss challenging topics to improve conflict negotiation and assertive communication and receive feedback from group members.

**Health promotion control.** We compared IMARA to a health promotion control program matched in length, intensity, and time spent in joint mother-daughter activities to control for non-specific therapeutic effects associated with contact time, group relationships, and facilitator attention. However, we ensured that the content of the program did not address sexual and reproductive health or mother-daughter communication, the active ingredients hypothesized to impact outcomes in IMARA. The health promotion program emphasizes healthy eating and nutrition, physical activity and exercise, informed consumer behavior, beauty standards for Black/African-American women and girls, personal and societal facts and costs of drug/alcohol use, and violence prevention. The curriculum is based on a health promotion program used in prior research [22], but adapted to ensure topics addressed important general health concerns in the target population–urban Black/African-American women and girls. Similar to IMARA, the health promotion curriculum employs an interactive approach, using games, videos, factual presentations by facilitators, and hands-on activities (e.g., planting herbs for nutritional benefits). Mothers and daughters learn relaxation exercises and how to understand nutritional labels. During joint activities, mother-daughter dyads discuss media images of beauty, watch videos, play games, and develop physical activity pyramids. In contrast to IMARA, mothers and daughters did not engage in role-plays or joint communication exercises, and they did not receive condom skills training. Specifically, IMARA participants learned the steps of condom use, practiced putting a condom on a penis model, and received feedback from facilitators and group members to ensure accuracy. By contrast, the health promotion participants watched a 10-minute video demonstrating condom use. The only overlap between IMARA and the health promotion content was a 15-minute game of Jeopardy to review the facts of STI/HIV transmission and prevention. We decided to provide STI/HIV information and a video of condom use given the high-risk profile of the population, although we recognize this may have attenuated group comparisons.

**Facilitator training.** Facilitators in both IMARA and the health promotion program were Black/African-American women with backgrounds in psychology, Bachelors or Master's degrees, previous experience leading groups and working with youth in health settings. Facilitators did not overlap arms and were trained separately. Facilitators in both arms were trained in groups over a month (30 hours total). Training was led by the project director and an experienced facilitator with graduate-level education. Where possible, facilitators also observed at least one full workshop prior to delivering the program, and in all cases, new facilitators were

paired with an experienced facilitator to lead their first two workshops. Training reinforced the importance of manualized interventions, following intervention content, and techniques to encourage girls' and mothers' participation. Facilitators practiced each session, conducting "mock run-throughs," and received extensive feedback until deemed competent by the project director. Training also included the facts of STI/HIV transmission and prevention, adolescent psychosexual development, group dynamics, and behavior management.

**IMARA fidelity.**   To evaluate treatment fidelity to the IMARA curriculum, facilitators rated their adherence to curricular activities at the end of each workshop. Ratings were reviewed by the project director following each workshop session and discussed during supervision along with any concerns and need for additional training. Annual refresher trainings were conducted to prevent facilitator drift from the curriculum. Treatment fidelity based on facilitator reports was 97%.

### Data analyses

Power analyses determined the sample size needed to identify differences in the presence of a new STI at 12-month follow-up among participants from IMARA versus the health promotion control group. Our registered, a-priori power calculations were conducted assuming a 1:1 randomization scheme: Using a two-sided test with alpha = .05, a sample of $n = 200$ would yield $\geq$ 80% power to detect a relative risk of $\leq 0.40$ for the two-group comparison, with new STI prevalence in the control group at 25%. Power was not substantially altered when assessed retrospectively for a 1.5:1 allocation favoring IMARA, as was ultimately arrived at in the present study.

Log-binomial regression was used to test differences between arms in the presence of a new STI at 12-month follow-up, adjusting for the presence of any STI at baseline. The new STI outcome at follow-up was a binary variable (yes/no to testing positive for at least 1 of the 3 STIs), because the study was not powered to examine specific STI outcomes. Standardized differences in means or proportions were calculated between arms for each baseline variable. All standardized differences were small ($< |.20|$), and thus, covariates were not included in the regression models, except for baseline STI status (yes/no). This exception was made for consistency with existing literature: past STI occurrence is strongly associated with future acquisition, and so it is standard to co-vary for STI history when predicting future infection. We conducted the log-binomial analyses in two ways. (1) Consistent with the original analytic plan registered at Clinicaltrials.gov, we ran the model among only those participants who provided all STI data, and for whom STI status could thus be known at both time points ($n = 164$). (2) We also conducted a follow-up intent-to-treat analysis in which participants with missing follow-up data were coded according to their baseline STI status. This approach, also known as the last observation coded forward method, provided a more conservative check on the original modeling. Robust variance estimation (sandwich estimator) accounted for correlations among participants within clusters (i.e., groups in which dyads participated) in all models. Note that log-binomial regression is appropriate for estimating the relative risk of a binary outcome that is not infrequent (e.g., occurring in $> 5\%$ of cases). All log binomial models were re-run including an interaction between arm and baseline STI status in order to evaluate if any intervention effects were moderated by baseline STI status. The interaction term was small and non-significant in each case, failing to provide support for significant moderating effects. We thus present findings from the models without the interaction term. All analyses were performed using SAS version 9.4.

### Results

Fig 1 shows participant flow. Of the 670 dyads screened, 142 (21%) were ineligible primarily due to the girl not meeting the age requirements and/or the family being unable to attend

**Table 1. Descriptive statistics in girls at baseline.**

| Variable | Full Sample | | |
| --- | --- | --- | --- |
| | Total ($n$ = 199) | IMARA ($n$ = 118) | Control ($n$ = 81) |
| | Mean (SD) | Mean (SD) | Mean (SD) |
| | or % ($n$) | or % ($n$) | or % ($n$) |
| **Sexual Risk** | | | |
| Positive for any STI | 19% (36) | 21% (24) | 15% (12) |
| Mean number of STIs | 0.21 (0.48) | 0.25 (0.52) | 0.16 (0.40) |
| Ever had sex | 38% (74) | 38% (44) | 38% (30) |
| Sexually active in past 6 months* | 74% (55) | 77% (34) | 70% (21) |
| Condom use at last sex* | 58% (43) | 59% (26) | 57% (17) |
| **Demographics** | | | |
| Age at baseline (years) | 16.02 (1.35) | 16.07 (1.41) | 15.95 (1.26) |
| Black/African-American race | 99% (197) | 99% (117) | 99% (80) |
| Hispanic ethnicity | 2% (3) | 0% (0) | 4% (3) |
| Mother lived on less than $30,000 | 78% (155) | 77% (91) | 79% (64) |

*Note.* All standardized differences between arms in the means or proportions for each variable were < |.20|. Possible number of STIs ranged from 0–3. Ever had sex indicated lifetime vaginal/anal sexual activity. Sexual activity in the past 6 months and condom use at last sex both referred to vaginal/anal sexual activity. Financial data were collected from mothers and reflected amount lived on over the past year. All percentages were calculated based on available data.

*Sexual activity in the past 6 months and condom use were assessed only in the subset of girls who reported ever having been sexually active: $n$ = 74 in the full sample, $n$ = 44 in IMARA, and $n$ = 30 in Control.

sessions. Of the 528 eligible families, 174 (33%) declined to participate, and 98 (18%) did not have working telephone numbers to schedule appointments. Thus, 256 dyads completed the baseline assessment. Consistent with previous research [20], attrition between baseline assessment and intervention start was 22% consisting of families no longer interested (2%) and families who were scheduled but did not show up (20%). This resulted in 199 dyads randomly assigned to IMARA or the health promotion program. Retention at 12-months was 84% for IMARA and 90% for the health promotion program and did not differ between arms (Fig 1).

Table 1 presents baseline descriptive statistics for the sample. Caregivers were biological mothers (64%), aunts (16%), grandmothers (11%), other (5%), and adoptive mothers (4%), ranging in age from 21–75 years (M = 42.82; SD = 10.36), and 78% reported family earnings <$30,000 annually. Over one-third of the girls (38%, $n$ = 74) reported ever having had vaginal/anal sex at baseline. Of these, 74% ($n$ = 55) reported vaginal/anal sex in the past six months, and 42% ($n$ = 31) indicated not using a condom at last vaginal/anal sex. Notably, within the subsample positive for a STI at baseline, only 66%, ($n$ = 23) indicated lifetime vaginal/anal sex. Of these, 70% ($n$ = 16) reported vaginal/anal sex in the past six months, and 43% ($n$ = 10) reported not using a condom at last vaginal/anal sex. Caregiver type was not significantly associated with testing STI-positive at baseline ($p$ >.05). Percentages reflect available data.

Regarding mental health, 26% and 27% of the sample self-reported clinically significant externalizing and internalizing symptoms on the Youth Self-Report [23] at baseline, respectively. These rates did not differ significantly between arms ($p$ >.05). For an in-depth summary of mental health symptoms in the sample, including the effects of IMARA on externalizing and internalizing symptom trajectories over 12 months and the associations of symptom change with the protective effect of IMARA against future STIs, see Kendall et al. [24].

As noted, this study was not powered to test the effects of IMARA on specific types of STIs. For descriptive purposes, however, we calculated the prevalence rates of each STI. At baseline among girls in IMARA, prevalence rates for chlamydia, gonorrhea, and trichomoniasis were

13% (*n* = 15), 3% (*n* = 3), and 9% (*n* = 10), respectively. At 12 months, these respective rates were 8% (*n* = 8), 3% (*n* = 3), and 7% (*n* = 7). Among girls assigned to the control arm, baseline rates of chlamydia, gonorrhea, and trichomoniasis were 9% (*n* = 7), 4% (*n* = 3), and 4% (*n* = 3). These respective rates at 12 months were 15% (*n* = 11), 4% (*n* = 3), and 11% (*n* = 8). Girls could test positive for more than 1 STI. Among girls who tested positive for at least 1 STI at baseline, 37% (*n* = 13) reported having ever been told by a medical provider that they were STI-positive; of those, 54% (*n* = 7) had been delivered this news in the past 6 months. At 1 year, 12% (*n* = 4) of girls who tested positive for a STI indicated they had been told by a provider outside of the study about their status. All percentages were calculated based on available data.

Girls attendance at both sessions of IMARA and the health promotion program was 80% (*n* = 160). Partial attendance did not differ across intervention arms; 24% in IMARA and 14% in health promotion program, $\chi^2$ (1, 198) = 2.78, *p* = .103. Reasons for non-attendance included scheduling conflicts, illness/hospitalization, childcare, and transportation/car problems. At 12-month follow-up, there were no significant differences in missing STI outcomes by intervention arm (19% IMARA vs. 11% health promotion; p = 0.122).

Prospective analyses tested the difference between conditions in a new STI at 12-months, adjusting within the same model for baseline STI status and cluster robust standard errors. Among girls who provided STI data at both time points (*n* = 164), the risk of a new STI at 12-months was 43% lower for girls from IMARA than the health promotion program, relative risk (RR) = .57, 95% confidence interval (CI) [0.37–0.88], *p* = .011. Within this same model, the effect of the baseline STI status covariate on STI acquisition by 12-months was RR = 3.24, 95% [CI 1.78–5.93], *p*<.001; that is, people who were STI-positive at baseline were much more likely to have an incident STI at follow-up. Among girls with observed data at each time point, STI rates in IMARA went from 19% (*n* = 18) at baseline to 16% (*n* = 15) at 12-months, and rates went from 14% (*n* = 10) to 24% (*n* = 17) in the health promotion program (Fig 2).

Next, we ran a follow-up intent-to-treat analysis with the last observation coded forward for *n* = 192 participants to provide a more conservative evaluation of treatment effects. Seven participants were excluded from this analysis, three participants with unconfirmed STI treatment at baseline and four participants who were not tested at baseline. This method also provided support for a significant protective effect of IMARA against future STIs, RR = 0.63, 95% [CI 0.43–0.91], *p* = .014. Within this model, the effect of the baseline STI covariate was RR = 4.39, 95% [CI 2.49–7.74], *p*<.001.

## Discussion

This study provides preliminary efficacy data indicating that IMARA protects against incident STIs at one-year follow-up among Black/African-American adolescent girls. In a 2-arm group RCT with conditions matched in time and intensity, girls from IMARA showed a 43% lower likelihood of a new STI compared to those in a time-matched mother-daughter health promotion control program. Given the preliminary nature of the data, and the potential for spurious effects, we followed up with a more conservative intent-to-treat analyses by assigning youth with missing outcome data the same STI status as their baseline. This approach also showed that IMARA was associated with a significantly lower likelihood of future STI acquisition compared with the control arm, albeit with a weakened magnitude (37%) compared with the original model. Findings advance previous research by producing strong effect sizes and treatment effects on biologically determined STIs over 12 months for Black/African-American adolescent girls, who are at elevated risk of STI/HIV relative to White and Latina peers [4, 25].

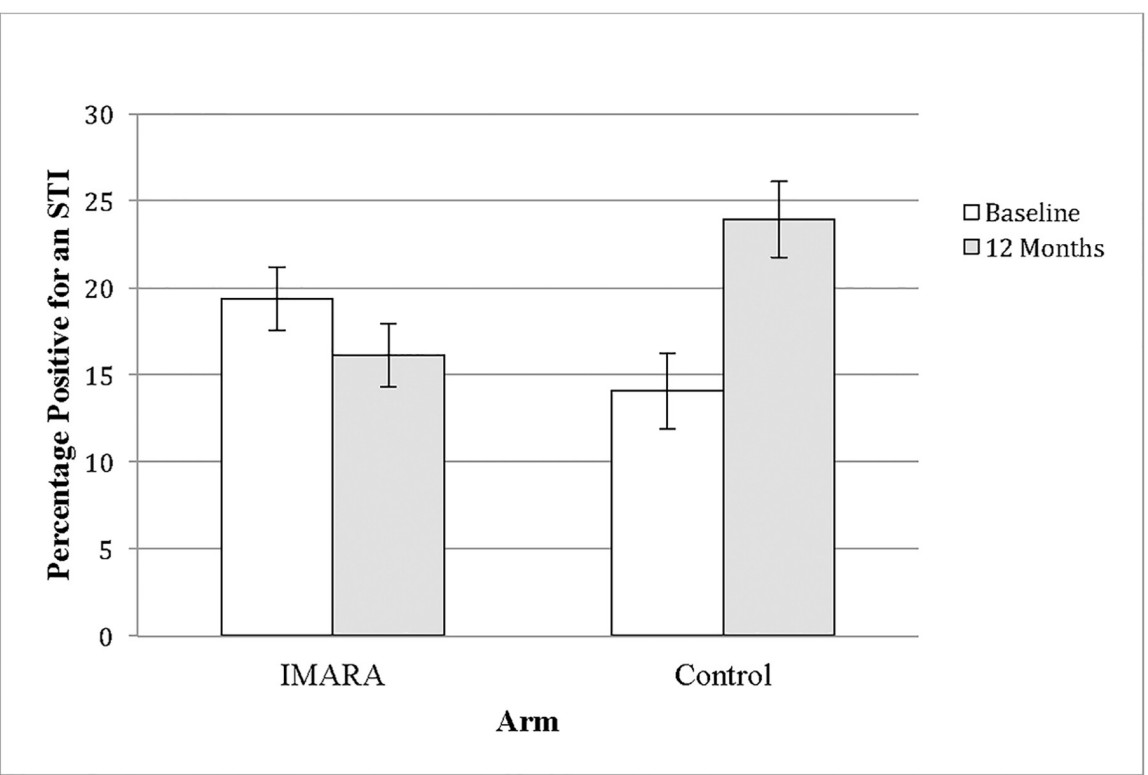

**Fig 2. STI prevalence rates at each time point.** STI prevalence rates among girls with observed data at both time points at baseline
(n = 18/93 in IMARA; n = 10/71 in the control group) and 12-month follow-up (n = 15/93 in IMARA; n = 17/71 in the control group).
Error bars indicate the standard error of the probability within a given arm.

Prior research has been limited by short-term effects (e.g., 3 or 6-month follow-up) [6, 16],
non-behavioral outcomes (e.g., attitudes, beliefs) [7], and self-reported sexual behavior [8, 9].
SISTA and Project STYLE did not report biologically determined STI outcomes [18, 20], the
primary outcome of this study, but they did demonstrate change in self-reported sexual behav-
ior. Like IMARA, girls who received SiHLE were less likely to acquire a new chlamydia infec-
tion over 12-months than girls in the comparison group. However, SiHLE was delivered over
four Saturdays, a potentially more burdensome schedule, compared to IMARA's two-day
workshop, and SiHLE did not include girls from mental health clinics–a uniquely challenging
and vulnerable population. Future research should compare the treatment effects and cost-
effectiveness of active HIV prevention programs to determine the best use of limited
resources.

Several factors may explain our positive effects, but future research must formally test medi-
ators. IMARA builds on a social-personal framework [17] and empirical findings [11, 13]
emphasizing the central role of mothers in daughters' sexual development and decision-mak-
ing. By focusing on the mother-daughter dyad, IMARA leverages an important social resource
for girls and contributes a new strategy to the STI/HIV prevention toolkit [26]. IMARA aims
to change girls' behavior in part by strengthening daughter's perceptions of mothers as a sup-
port system in sexual decision-making, and by shifting mothers' and daughters' peer norms
within the group in favor of prevention. IMARA's focus on building relationships may be par-
ticularly salient for adolescent girls as increased technology is minimizing human connections.

IMARA builds on previous findings that strong mother-daughter communication is related to less adolescent risky sexual behavior, [11, 12, 14] with facilitators teaching dyads to communicate effectively about sexual topics. In doing so, IMARA may set the stage for continued conversations as girls reach new developmental milestones. This may explain why IMARA's positive effects were observed out to one year after the workshops; testing if improvements in mother-daughter communication mediate IMARA's intervention effects is important for future inquiry.

Finally, IMARA targets individual and social mechanisms shown to affect sexual behavior and these may help explain the reduction in STIs. For example, mental health [27], greater relationship power [28, 29], increased ethnic and gender pride [30], and stronger emotion regulation [31] may influence STI outcomes. IMARA promotes mothers as role models for daughters, demonstrating pride in womanhood and Black/African-American identity, cautioning girls about the imbalance of power in age-discrepant relationships, and supporting emotion regulation strategies that improve safe-sex decisions. Indeed, recent work from our group using the same sample as in the present study showed that among girls who entered with high versus lower externalizing mental health symptoms, those who received IMARA showed a significantly greater decrease in externalizing scores relative to the control group [24]. Furthermore, among these girls, symptom improvements appeared to be associated with IMARA's protective effect against new STIs, suggesting that mental health change might mediate the intervention effects. Future study should formally evaluate the role of mediators of IMARA's positive STI treatment outcomes.

Notably, 34% of STI-positive girls at baseline reported never having had vaginal/anal sex. This is consistent with other studies [32]. It is possible that girls did not understand the questions, were unable to recall prior sexual experiences, were reluctant to report sexual activity because their caregiver was involved in the study, or had a narrow definition of sexual behavior that excluded certain activities (e.g., anal sex). Likewise, girls may not have endorsed unwanted sex or simply preferred not to disclose. Findings underscore the unique value of biological outcomes relative to self-reports. Future studies should seek to understand the discrepancy between objective and self-reported data and identify ways to improve the validity of self-reports, perhaps through integration with biological markers [32]. Diminished socioeconomic resources heighten risk for STIs and HIV [33, 34], yet IMARA appears to have protected against STIs in girls despite these structural challenges, as 78% of mothers reported < $30,000 annual family income, and 75% indicated being single parents.

Study limitations should be considered in light of these promising effects. This study represents a single test of IMARA; replication is required. Mothers and daughters were not blinded to condition because intervention content revealed assignment status. However, the interventions were matched in length, intensity, and amount of time mothers and daughters spent together to control for these non-specific factors. Results pertain to Black/African-American adolescent girls and their mothers and may not generalize to other groups. Patterns may differ depending on the type of caregiver. Future studies with greater power could perform a stratified analysis to unpack if caregiver type (e.g., biological mother, aunt) is related to the intervention effects. As with many efficacy trials where participants are compensated to participate in an intervention, this study reimbursed mothers and daughters each $40 to attend each 6-hour workshop day. Providing compensation may limit the feasibility of participation outside the research study, but the small amount–about $7 per hour–was designed to be non-coercive and offset the potential for lost wages. Observed differences in STI rates between groups at one year may suggest girls in IMARA were more likely to be tested and treated during the 12-months following the intervention. This alternative interpretation, however, still supports IMARA's benefits since testing and treatment are themselves important public health

outcomes. We did not test participants for HIV, because of the low incidence among adolescent girls. However, STIs are a proxy for unprotected sexual activity, a known risk for HIV. Despite limitations, the study has several strengths, namely use of biological STI outcomes, long-term follow-up, and strong effect sizes.

Few STI/HIV evidence-based interventions for Black/African-American adolescent girls translate into "real-world" settings. Next steps will require careful attention to the factors influencing IMARA's reach, adoption, implementation, maintenance and dissemination [35]. IMARA's length and number of facilitators may present barriers to dissemination and implementation, particularly in resource-poor environments. However, by delivering IMARA over two days, we sought to lessen family burden from more typical multi-session weekly interventions. Moreover, as a first test of IMARA, the evidence of effectiveness in this study can serve as a springboard for future adaptations for specific settings and contexts. Two examples are currently underway; we are exploring the feasibility and effectiveness of a one-day workshop, and we are adapting IMARA for the South African context. Partnering with setting-specific stakeholders will drive adaptation (e.g., length, content), delivery (e.g., format, number of facilitators), and evaluation. Additionally, adaptive designs [36] can test and distinguish the potent components of IMARA to create a streamlined curriculum. Likewise, evaluating alternative implementation approaches (e.g., embedding IMARA in sexual health services, schools, faith-based organizations) can inform optimal delivery. Finally, IMARA can be adapted to include new prevention technologies to alleviate STI/HIV stigma and improve prevention behavior.

## Supporting information

**S1 Dataset.**
(XLSX)

**S1 File.**
(DOC)

**S2 File. CONSORT 2010 checklist of information to include when reporting a randomised trial**∗**.**
(DOC)

## Acknowledgments

We thank our collaborating partners and institutions in the conduct of the study. We are especially grateful to the girls and their primary caregivers who trusted us and participated in the program. Faith Fletcher is located at the Department of Health Behavior, School of Public Health, University of Alabama at Birmingham.

## Author Contributions

**Conceptualization:** Geri R. Donenberg, Erin Emerson.

**Data curation:** Erin Emerson.

**Formal analysis:** Ashley D. Kendall, Bethany C. Bray.

**Funding acquisition:** Geri R. Donenberg.

**Methodology:** Geri R. Donenberg, Erin Emerson.

**Project administration:** Kelly McCabe.

**Supervision:** Geri R. Donenberg, Erin Emerson, Kelly McCabe.

**Visualization:** Ashley D. Kendall.

**Writing – original draft:** Geri R. Donenberg, Ashley D. Kendall.

**Writing – review & editing:** Erin Emerson, Faith E. Fletcher, Bethany C. Bray, Kelly McCabe.

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
