## [Decision Letter · Decision Letter 0]

23 Jan 2020

PONE-D-19-32194

IMARA: A mother-daughter group randomized controlled trial to reduce sexually transmitted infections in Black/African-American adolescents

PLOS ONE

Dear Dr. Donenberg,

Thank you for submitting your manuscript to PLOS ONE. After careful consideration, we feel that it has merit but does not fully meet PLOS ONE’s publication criteria as it currently stands. Therefore, we invite you to submit a revised version of the manuscript that addresses the points raised during the review process.

You will see that the Referees found your work of some interest. However, they also raised major criticisms. Please respond to all the comments by Reviewers, with special attention to methodological points raised by Reviewer #3.

We would appreciate receiving your revised manuscript by April 30 2020. To enhance the reproducibility of your results, we recommend that if applicable you deposit your laboratory protocols in protocols.io, where a protocol can be assigned its own identifier (DOI) such that it can be cited independently in the future. For instructions see: http://journals.plos.org/plosone/s/submission-guidelines#loc-laboratory-protocols

We look forward to receiving your revised manuscript.

Kind regards,

Giuseppe Vittorio De Socio, MD, PhD

Academic Editor

PLOS ONE

Journal Requirements:

2. We note that you have reported significance probabilities of 0 in places. Since p=0 is not strictly possible, please correct this to a more appropriate limit, eg 'p<0.0001'.

3.  Thank you for submitting your clinical trial to PLOS ONE and for providing the name of the registry and the registration number. The information in the registry entry suggests that your trial was registered after patient recruitment began. PLOS ONE strongly encourages authors to register all trials before recruiting the first participant in a study.

i) your reasons for your delay in registering this study (after enrolment of participants started);

ii) confirmation that all related trials are registered by stating: “The authors confirm that all ongoing and related trials for this drug/intervention are registered”.

Please also ensure you report the date at which the ethics committee approved the study as well as the complete date range for patient recruitment and follow-up in the Methods section of your manuscript.

Reviewers' comments:

Reviewer's Responses to Questions

**Comments to the Author**

1. Is the manuscript technically sound, and do the data support the conclusions?

Reviewer #1: Yes

Reviewer #2: Partly

Reviewer #3: Partly

2. Has the statistical analysis been performed appropriately and rigorously? 

Reviewer #1: Yes

Reviewer #2: Yes

Reviewer #3: No

3. Have the authors made all data underlying the findings in their manuscript fully available?

Reviewer #1: No

Reviewer #2: Yes

Reviewer #3: Yes

4. Is the manuscript presented in an intelligible fashion and written in standard English?

Reviewer #1: Yes

Reviewer #2: Yes

Reviewer #3: Yes

5. Review Comments to the Author

Reviewer #1: The authors conducted a randomized controlled trial to reduce sexually transmitted infections in Black/African –American adolescents. I found the paper to be written well with a clear logical flow and tight focus. However, I have identified minor concerns that the authors should work on.

Methods Line 97-99,page 5 Why was the mental health clinic targeted for the initial enrollments? Why this sampling strategy? This may suggest that the study team was more interested in a vulnerable population who definitely may have relational issues with their mothers/caretakers. This could potentially bias the effect of the intervention making it appear positive when it is not. Are you able to describe what mental health problems they had to assure the readers that this would not bias your findings?

Line 113, page 6 What do you mean by dyads learned their group assignment after the baseline assessment....? Make this sentence clear to the reader.

It is not clear whether the curricular used were hard copies or soft copies on tablets etc. Relatedly, the inclusion criteria ....spoke English.... do the interventions need one to speak English only without reading? Did you do any language proficiency testing for these adolescents?

Line 141, noted rationale for not doing HIV testing but given the strong association between HIV transmission and STIs; at the subset of adolescents that had STIs at 12 months should have received HIV screening. Are the authors able to get this data and include it? may be from a related program etc

Line 143-144,page 7 consistent with Illinois law, expedited partner therapy was offered. Please provide a citation or describe what procedures are done to ensure partner was offered therapy. Also make clear for how long treatment for the STIs was?

Line 164, add a citation in support of this information.

Line 165-166, page 8- was this work re acceptability and feasibility published?if yes provide a citation.

Line 180 Mothers and daughters separately review..... is this referring to the mother and daughter dyad? please make clear

Line 203 - .... and they did not receive condom skills training. I find this quite surprising in this day and age. Line 205 refers to a 10 minute video demonstrating condom use. How different is the content of this video and the condom skills training? Please clarify

IMARA Fidelity

Noted facilitators rated their adherence to curricular activities. What was fidelity based on the directors report?

Results

Did the STI risk differ with type of caregiver?

Lines 275-279- State the reasons for non attendance as they may be related to the intervention thus affecting feasibility.

Describe what STIs were identified at baseline and at 12 months. Do they differ? Were there any negative outcomes from the intervention e.g caretaker violence, depression, daughter withdrawal etc

Discussion

Line 312 Describe what you mean by advance previous research? may need to reword

Line 314, describe the short term effects and non behavioral outcomes. How do they differ from what the IMARA intervention resulted into?

Line 339- treatment outcomes - clarify which treatment (IMARA or the STI treatment)

Line 360.... in IMARA were more likely to be tested and treated during the 12 months following intervention. This introduces surveillance bias. Can the authors describe how this bias was overcome as it affects the main study outcome.

References- Have the citations in brackets as per PLOSONE reference style.

Reviewer #2: This paper presents the results of a randomized controlled trial that evaluates an intervention for African-American teenage girls and their mothers/caretakers. The strengths of the research include a clearly described intervention (and control), good number of study participants, with excellent follow-up rates. This much have been a very challenging study to set up, conduct and implement. The limitations of the study include apparently contradictory findings in crude vs. adjusted analysis, and limited information on any outcome other than STI rate itself.

Specific comments follow.

Overall. The introduction is good. A line or two could be added to describe the potential “mechanisms” by which this intervention might reduce STIs – would it be due to less sex, increased condom use, or lower risk partners?

Can the authors include the specific trial number for clinicaltrials.gov?

Could be helpful to understand how much participants were paid to attend the workshops, to help understand its feasibility outside of a research study.

It would be helpful to know what diagnostic test kit was used to test for chlamydia, gonorrhea and trichomonas (and whether there could be false positives).

Results –

Can the authors present which specific infections were actually diagnosed – did this intervention reduce chlamydia and gonorrhea and trichomonas? – or was most of the effect related to trichomonas?

Were there any questions about self-reported STIs diagnosed outside of the study between baseline and follow-up? Or questions about possible mediators such as self-reported sexual behavior at 12-month followup? Or mother-daughter discussions about sexual behavior?

It remains a bit unclear what conclusion to make when the unadjusted RR was 0.59 (lower risk), and the adjusted RR was 3.2 (higher risk when comparing to baseline status). This conflicting finding should be discussed more in the discussion. If there was an “a-priori plan” to adjust for baseline STIs included in the clinicaltrials.gov protocol, then the authors should probably be focusing on that result (which showed an increased risk in the intervention group).

Minor issues –

Methods – the description of the randomization could be more clear how more people were randomized to the intervention – and could refer to the consort diagram (optional – this is presented in the results but could be in the methods).

Minor – line 246 clarify if the “treatment” refers to the participation in the intervention (or some other sort of treatment…)

Minor – Table 1 – more clarify the denominators for the percentages for “sexually active last 6 months and condom use” (mentioned in the footnote, but not obvious when first reading through the table).

Reviewer #3: This is an important randomised controlled trial testing the efficacy of the IMARA project (a mother-daughter psycho-social STI/HIV prevention program) vs. a control arm of a more generic intervention promoting healthy eating and nutrition, physical activity and exercise, in a group of 199 adolescent Black/African-American girls’ incident STIs at 12 months.

This study provides preliminary efficacy data indicating that IMARA did protect against the primary endpoint of incident STIs at one-year follow-up.

However, there are aspects of the study design and analysis that could have confounded the results and which need to be clarified before drawing firm conclusions on the efficacy of this intervention. This is key before implementing similar program in other target populations.

Main points

1. Randomisation schedule was created in an unusual way. Dyads were asked to select their randomisation treatment from a paper bag. This could have led to systematic bias and Table 1 indeed shows large imbalances for some of the participants’ characteristics at baseline (e.g. % positive for STI, mean number of STI, ethnicity, history of sex and socio-economic status). These are key common causes of both treatment allocation and incidence of STI so could have confounded the results. Authors should re-perform the analysis after controlling for all these additional imbalanced factors not just for the presence of STI at baseline. Residual confounding should be also mentioned in the limitations (lines 356 onward).

2. Similarly, the procedure of alternating between IMARA and the health promotion program would have made the process predictable and broke down the concealment of allocation so that recruiters could have decided not to enrol, say, a problematic dyad if it was known that next allocation would have been for example control and not IMARA.

3. Mothers and daughters in the control arm did not engage in role-plays or joint commune exercises, resulting in lack of blinding. Unclear why efforts were not made to try to implement blinding. This is an additional limitation that should be mentioned in the Discussion.

4. Calculation of statistical power is cryptic. It is said that dyads were randomly assigned to the IMARA (n=118) or the health promotion control program (n=81) following an overall ratio of roughly 1.5:1 favouring IMARA to enhance analytic power to detect the treatment effect. Nevertheless, in the Methods power is given for a total sample size of N=200 participants, assuming an even split of 100 vs. 100 in the 2 groups. Authors should show the difference in power using the 1:1 vs. the 1:1.5 random allocation and explicit reasons for choosing the latter. What was the expected increase in power? In general, from the text in lines 251-258, it appears that the sample size of 199 resulted as a consequence of an inevitable attrition between baseline assessment and initiation of the intervention rather than a pre-planned design.

5. Although similar by treatment arm, the rate of non-adherence/loss to follow-up was not negligible at 16% and 10% in the two arms. Because of that it is possible that the two groups are no longer exchangeable at 12 months. It is not an unreasonable assumption in these settings that daughters who are lost to follow-up are more likely to be have acquired STIs. Authors should perform an intention to treat analysis using the missing-failure approach. Alternatively, it is possible that the effect of the strategy could have been diluted by attrition so authors should also perform an on-treatment analysis after controlling for post-baseline confounding factors using a marginal model. This should control for all possible post-baseline prognostic factors as well as common causes of acquiring STI and risk of drop-out.

6. Lines 264-273. The issue of baseline exchangeability between arms is ruled out in the Methods with a couple of sentences saying that there was no difference at baseline between the two groups. A p-value >0.05 for these comparisons (line 236) is meaningless and should not be reported as this p-value should be equal to 1 as the null hypothesis is true by definition in a trial. On the other hand, there are larger imbalances between baseline factors which should be described in the text and controlled for in the analysis. For example, mothers in the IMARA arm were less likely to earn <$30k per annum that mothers in the control arm and this could have explained the difference in incidence of STI at 12 months. The sentence that overall >75% (which becomes ‘almost 80%’ in line 351 of Discussion section) of mothers earned <$30k annually in both groups is inaccurate and misleading.

7. Line 288. This same model adjusted for baseline STI status was RR=3.24, 95% [CI 1.72-6.11], p=.000. Has the direction of RR being inverted? Is this the difference of control vs. IMARA?

8. The fact that the difference observed could be due to mediators is mentioned and several potential mediators are listed e.g. improvements in mother-daughter communication etc. Was use of condom measured in the study? Change in sexual risk behaviour seems to be the most obvious surrogate endpoint/mediator in this analysis to verify.

9. Line 359. American adolescent girls and their mothers and may not generalise to other groups. Patterns may differ depending on the type of caregiver. For how many dyads the caregiver was not the natural mother? Unclear why a stratified analysis by type of caregiver (mother vs. other) could not be performed.

10. What was the effect of SISTA SiHLE and STYLE compared to this intervention? Results of the various trials should be reported and compared in the Discussion section.

Other points

1. Lines 248-250. This stringent approach was not intended to be fully powered, but rather to provide a preliminary check of whether the pattern of results observed within the full sample replicated within the subset known to be sexually active at baseline. Indeed, it is very likely to actually be under-powered. Would have been more sensible to restrict the analysis to those who did not show STI at baseline?

2. Line 306. Suggested rewording: This study provides preliminary efficacy data indicating that IMARA protects against incident….

3. Line 308. The authors claim that for consistency with existing literature past STI occurrence is strongly associated with future acquisition, and so it is standard to co-vary for STI history when predicting future infection. Nevertheless, the unadjusted RR of 0.43 is mentioned here, why?

4. Line 341. Suggested rewording: It is possible THAT girls did not understand the….

If the data are consistent with the literature, why is paragraph (in lines 341-349) needed at all? Or is it instead the case that 33% is likely to be an under-estimate of the prevalence of STI at baseline?

6. PLOS authors have the option to publish the peer review history of their article (what does this mean?). If published, this will include your full peer review and any attached files.

Reviewer #1: No

Reviewer #2: No

Reviewer #3: No

---

## [Author Response · Author response to Decision Letter 0]

21 Apr 2020

April 20, 2020

Giuseppe Vittorio De Socio, MD, PhD

Academic Editor

PLOS ONE

Dear Dr. Vittorio De Socio:

Re: IMARA: A mother-daughter group randomized controlled trial to reduce sexually transmitted infections in Black/African-American adolescents (PONE-D-19-32194)

Thank you for the opportunity to revise and resubmit for review the above referenced manuscript. We appreciate the very thoughtful and helpful comments from prior reviewers, and we believe that in responding to them, the paper is significantly improved. 

In order to address the reviewer’s analytic concerns, we sought the expertise of Dr. Bethany Bray. The original data analyst, Dr. Anna Hotton, was no longer available, and thus, we have updated the authorship list and order to accurately represent the contributions to the manuscript.

Please see each reviewer comment in bold and our response in regular text. Changes in the manuscript are indicated by page and line numbers and they are tracked in the submitted manuscript. We sincerely hope these changes meet with your and the reviewers’ approval. 

Please note that we have now included our clinical trial protocol as Supporting Information file.

Sincerely,

Geri R. Donenberg, Ph.D.

Professor of Medicine and Psychology ⏐ University of Illinois at Chicago

Co-Vice Chair of Research ⏐ Department of Medicine

Director ⏐ Healthy Youths Program 

Director ⏐ Center for Dissemination and Implementation Science 

Editors Comments and Journal Requirements

i. Please ensure that your manuscript meets PLOS ONE's style requirements, including those for file naming.

We reviewed the manuscript for proper labeling and style requirements.

ii. We note that you have reported significance probabilities of 0 in places. Since p=0 is not strictly possible, please correct this to a more appropriate limit, eg 'p<0.0001'.

We have corrected p = .000 to read p < .001. 

iii. … “The information in the registry entry suggests that your trial was registered after patient recruitment began. PLOS ONE strongly encourages authors to register all trials before recruiting the first participant in a study. As per the journal’s editorial policy, please include in the Methods section of your paper”:

iiia) your reasons for your delay in registering this study (after enrolment of participants started);

As now described on p. 6-7, lines 167-195. “This trial is registered at clinicaltrials.gov (NCT02958813). Registration was delayed until 1½ months following the publication of the 2016 Final Policy on the Dissemination of NIH-funded Clinical Trial Information. Up to that point, NIH-funded studies were not required to register, and it was unclear whether the current behavioral study qualified because it did not involve experimental drugs. The Final Rule’s publication expanded and clarified registration requirements, thereby prompting registration.”

iiib) confirmation that all related trials are registered by stating: “The authors confirm that all ongoing and related trials for this drug/intervention are registered”.

This statement has been added to the methods section on p. 6, lines 195-196. 

Please also ensure you report the date at which the ethics committee approved the study as well as the complete date range for patient recruitment and follow-up in the Methods section of your manuscript.

The data of ethics committee approval is provided on p. 5, line 167. The complete date range for participant recruitment and follow-up is provided on p. 5, line 123.

iv. We note that you have indicated that data from this study are available upon request. PLOS only allows data to be available upon request if there are legal or ethical restrictions on sharing data publicly. 

At the time of original submission, we had not fully de-identified the data or obtained ethics approval to share the dataset. We agree to share a minimal de-identified data set once ethics approval is granted. This will be uploaded as a Supporting Information file.

Reviewer #1 

1. Methods Line 97-99, page 5. Why was the mental health clinic targeted for the initial enrollments? Why this sampling strategy? This may suggest that the study team was more interested in a vulnerable population who definitely may have relational issues with their mothers/caretakers. This could potentially bias the effect of the intervention making it appear positive when it is not. Are you able to describe what mental health problems they had to assure the readers that this would not bias your findings?

We initially targeted youth with mental health problems because we wanted to evaluate the intervention for families likely to have troubled mother-daughter relationships as the intervention was designed to specifically improve mother-daughter relationships. However, as noted by the reviewer, we did not want the intervention to be solely relevant to a vulnerable population, and thus, we opened recruitment to a more generalizable sample to provide a more robust evaluation of IMARA’s efficacy. In fact, only about one-quarter of the girls in the sample met criteria for clinically significant externalizing and internalizing symptoms, respectively, and these rates are similar to general population rates of non-mental health seeking teens. Moreover, the rates did not differ between treatment arms (p > .05). We have added this information to the manuscript to help assure readers that mental health problems would not bias our findings. We also now refer readers to a recent publication from our group that provides an in-depth summary of mental health symptoms in the sample, including testing the effects of IMARA on externalizing and internalizing symptom trajectories over 1 year and the associations of symptom change with future STI acquisition (Kendall et al., in press, Journal of Consulting and Clinical Psychology). Please see p. 14, lines 420-425.

2. Line 113, page 6. What do you mean by dyads learned their group assignment after the baseline assessment....? Make this sentence clear to the reader.

This has been clarified on p. 6, lines 146-149. 

3. It is not clear whether the curricular used were hard copies or soft copies on tablets etc. Relatedly, the inclusion criteria ....spoke English.... do the interventions need one to speak English only without reading? Did you do any language proficiency testing for these adolescents?

The curriculum was not delivered through written materials, but rather was conducted by two trained co-facilitators who themselves followed a written manual during in-person face-to-face groups. Being able to read, therefore, was not required to participate. English was an inclusion criterion because the curriculum was conducted in English by the facilitators. We now clarify that the intervention was delivered in English (see p. 7, lines 201 and 204).

4. Line 141. noted rationale for not doing HIV testing but given the strong association between HIV transmission and STIs; at the subset of adolescents that had STIs at 12 months should have received HIV screening. Are the authors able to get this data and include it? may be from a related program etc

We did not have sufficient resources to provide HIV testing to the full sample. However, adolescents who tested positive for a STI and received STI treatment from the study physician were offered HIV testing and partner notification services consistent with Illinois law. Still, these activities were beyond the research protocol, and we did not have participant consent to access this private health information. 

5. Line 143-144, page 7. Consistent with Illinois law, expedited partner therapy was offered. Please provide a citation or describe what procedures are done to ensure partner was offered therapy. Also make clear for how long treatment for the STIs was?

STI treatment was offered by the study physician, or youth could opt to receive treatment from their own physician. Where they received treatment from their own physician, we have no information about the partner notification process other than verification via a prescription from the doctor that STI treatment was provided. The standard treatment for all three STIs is single-dose directly observed therapy. We now clarify that STI treatment entailed single-dose directly observed therapy in the revised manuscript (see p. 8, line 221).

6. Line 164. add a citation in support of this information.

Two citations were added to confirm that SISTA and SiHLe contain content that addresses gender and ethnic pride (see p. 9, line 248).

7. Line 165-166, page 8. was this work re acceptability and feasibility published? if yes provide a citation.

This work has not been published.

8. Line 180. Mothers and daughters separately review..... is this referring to the mother and daughter dyad? please make clear

Some of the workshop activities occurred with the mothers as a group separate from daughters, and some of the activities were done with mothers and daughters together in the same group. This is clarified in the text (see p. 9, lines 259-277).

9. Line 203. .... and they did not receive condom skills training. I find this quite surprising in this day and age. Line 205 refers to a 10 minute video demonstrating condom use. How different is the content of this video and the condom skills training? Please clarify

In the IMARA intervention, mothers and daughters were taught the specific steps of condom use and practiced putting a condom on a penis model. They were provided this type of “hands-on” experience with feedback from the facilitators and the group members. By contrast, in the health promotion program, they watched a video that reviewed the steps of condom use, but they did not go through the steps or practice putting a condom on a penis model. This is clarified in the text (see p. 10, lines 296-305).

10. IMARA Fidelity. Noted facilitators rated their adherence to curricular activities. What was fidelity based on the directors report?

All facilitators met initial competency in intervention delivery following the training. Ongoing fidelity was evaluated using facilitator reports of adherence to specific activities. Data are not available regarding adherence as reported by the project director. However, the project director reviewed facilitator reports of adherence following each session and addressed any concerns during weekly meetings. In addition, annual refresher trainings were conducted to prevent facilitator drift from the curriculum. This is now noted in the manuscript (see pp. 11-12, lines 326-331).

Results

11. Did the STI risk differ with type of caregiver?

No, testing positive for an STI at baseline was not significantly associated with type of caregiver. We now report this on p. 14, lines 418-419. 

12. Lines 275-279. State the reasons for non-attendance as they may be related to the intervention thus affecting feasibility.

Reasons for non-attendance on day 2 included scheduling conflicts, illness/hospitalization, childcare, and transportation. These are now provided on p. 16, lines 462-463.

13. Describe what STIs were identified at baseline and at 12 months. Do they differ? 

We now describe STI frequencies by type (chlamydia, gonorrhea, trichomoniasis) in each arm at each time point (see pp. 14-15, lines 427-440). There were no significant differences between arms in the prevalence of any STI type at any time point (all p-values >.05). However, as noted in the text, the present study was not powered to detect the effects of IMARA on specific types of STIs. Thus, the STI frequencies are provided only for descriptive purposes. 

14. Were there any negative outcomes from the intervention e.g caretaker violence, depression, daughter withdrawal etc

In a recent publication now referenced in our manuscript (Kendall et al., Journal of Consulting and Clinical Psychology, in press), we conducted an in-depth analysis of the effects of the intervention on mental health symptoms over 12 months. We found that among girls who entered the study with high versus lower externalizing symptoms, those who received IMARA showed a slightly greater decrease in externalizing scores relative to the control (p = .035). Treatment was not associated with internalizing symptom change (p >.05). The study thus did not provide any evidence of negative mental health outcomes from the intervention. 

Two additional factors suggest no negative outcomes related to the study. First, the strong retention rate suggests that dyads felt positive returning for study visits. Second, each mother and daughter separately completed evaluations of the workshop at the end of both days and evaluations were routinely positive; none indicated negative outcomes. 

Discussion

15. Line 312. Describe what you mean by advance previous research? may need to reword

Thank you for the opportunity to provide further clarification. We re-worded the sentence as follows: “Findings advance previous research by producing strong effect sizes and treatment effects on biologically determined STIs over 12 months for Black/African-American adolescent girls, who are at elevated risk of STI/HIV relative to White and Latina peers [1, 2].” (see p.17, lines 552-555). 

16. Line 314. describe the short term effects and nonbehavioral outcomes. How do they differ from what the IMARA intervention resulted into?

IMARA demonstrated sustained outcomes at 12-months, rather than most studies which report outcomes immediately following an intervention, or at 3- and 6-months follow up. Similarly, “non-behavioral” refers to attitudes and beliefs which may be subject to social desirability bias. These definitions are now added to the sentence (see p. 17, lines 556-557).

17. Line 339. treatment outcomes - clarify which treatment (IMARA or the STI treatment)

We added “IMARA’s positive STI treatment outcomes.” (see p. 19, lines 628-629).

18. Line 360. .... in IMARA were more likely to be tested and treated during the 12 months following intervention. This introduces surveillance bias. Can the authors describe how this bias was overcome as it affects the main study outcome.

It appears the reviewer may have misread the sentence to be that “IMARA girls WERE more likely to be tested and treated.” The statement actually says: “Observed differences in STI rates between groups at one year may suggest girls in IMARA were more likely to be tested and treated during the 12-months following the intervention.” 

We do not know if this was the case and have no way to know. Rather, we sought to clarify that if this had occurred and explained the improvement in IMARA STI rates, it would actually provide additional evidence of positive effects of IMARA. However, if this is confusing, we would be happy to delete this sentence.

19. References. Have the citations in brackets as per PLOSONE reference style.

This has been done.

Reviewer #2 

1. Overall. A line or two could be added to describe the potential “mechanisms” by which this intervention might reduce STIs – would it be due to less sex, increased condom use, or lower risk partners?

We added to the statement in the Discussion proposing possible mechanisms related to STI outcomes as follows (see p. 18, lines 601-604): “Finally, IMARA targets individual and social mechanisms shown to affect sexual behavior and these may help explain the reduction in STIs. For example, mental health [3], greater relationship power [4, 5], increased ethnic and gender pride [6], and stronger emotion regulation [7] may influence STI outcomes.” 

2. Can the authors include the specific trial number for clinicaltrials.gov?

The trial number, NCT02958813, is now provided (see p. 6, line 167).

3. Could be helpful to understand how much participants were paid to attend the workshops, to help understand its feasibility outside of a research study.

We now clarify the payment for workshops in the Discussion and note it as a limitation with regard to feasibility (see p. 20, lines 664-668): “As with many efficacy trials where participants are compensated to participate in an intervention, this study reimbursed mothers and daughters each $40 to attend each 6-hour workshop day. Providing compensation may limit the feasibility of participation outside the research study, but the small amount – about $7 per hour – was designed to be non-coercive and offset the potential for lost wages.”

4. It would be helpful to know what diagnostic test kit was used to test for chlamydia, gonorrhea and trichomonas (and whether there could be false positives).

For chlamydia and gonorrhea, the name of the test is Abbott Real Time CT/NG assay.

CT sensitivity = 95.2% and specificity = 99.3%

NG sensitivity = 97.5% and specificity = 99.7%

For trichomonas, we used Taq-Man PCR. 

Emory Laboratory developed and validated this test and the sensitivity/specificity of the assay is 100% and 99.6% respectively. We would be happy to add this to the manuscript if the reviewer and editor would like.

Results –

5. Can the authors present which specific infections were actually diagnosed – did this intervention reduce chlamydia and gonorrhea and trichomonas? – or was most of the effect related to trichomonas?

Please see our response above to Reviewer 1, Point 14. 

6. Were there any questions about self-reported STIs diagnosed outside of the study between baseline and follow-up? 

Yes, youth reported at baseline and at 12-month follow-up on whether they had been told by a medical provider that they ever had an STI (at baseline) and over the past 6 months (at baseline and 12 months). We now report these descriptive statistics on p. 15, lines 442-446.

7. Or questions about possible mediators such as self-reported sexual behavior at 12-month follow-up? Or mother-daughter discussions about sexual behavior?

Participants completed a number of survey measures that focused on potential mediators of STIs. However, the primary outcome listed in clinicaltrials.gov was biologically determined STIs. We elected to focus on this outcome given our a priori hypotheses. That said, we do have data on mental health and mother-daughter communication. We have published an analysis of the mental health data, including the associations of mental health symptom change with the protective effect of the intervention against incident STIs (now mentioned in the manuscript; Kendall et al., in press, Journal of Consulting and Clinical Psychology). We plan to examine mother-daughter communication as a mediator of STI treatment effects in the future. Similarly, we have data on self-reported condom use but prefer to report this as a potential mediator after the primary outcome paper is published.

8. It remains a bit unclear what conclusion to make when the unadjusted RR was 0.59 (lower risk), and the adjusted RR was 3.2 (higher risk when comparing to baseline status). This conflicting finding should be discussed more in the discussion. If there was an “a-priori plan” to adjust for baseline STIs included in the clinicaltrials.gov protocol, then the authors should probably be focusing on that result (which showed an increased risk in the intervention group).

Thank you for the opportunity to clarify our writing, which we have done on p. 16. Both RRs were from the same, single model: the first (RR = .57 with 95% CI 0.37-0.88, p=.011) referred to the effect of arm (IMARA vs. health promotion control) on the 12-month STI outcome, and the second (RR = 3.24 with 95% CI 1.78-5.93, p<.001) referred to the effect of baseline STI status on the 12-month STI outcome. We hope it is now clear that there are no conflicting findings, as we are only reporting on the adjusted RR (along with the RR for the covariate), consistent with our a-priori analytic plan. 

Minor issues –

9. Methods. the description of the randomization could be more clear how more people were randomized to the intervention – and could refer to the consort diagram (optional – this is presented in the results but could be in the methods).

We have revised the description of randomization and now refer to the CONSORT diagram directly with the hope it is clearer (see p. 6, lines 146-161).

10. Minor – line 246. clarify if the “treatment” refers to the participation in the intervention (or some other sort of treatment…)

We clarify that “treatment” refers to the interventions.

11. Minor – Table 1. more clarify the denominators for the percentages for “sexually active last 6 months and condom use” (mentioned in the footnote, but not obvious when first reading through the table).

We now clarify in the footnote the denominators for the “Sexually active in past 6 months” and “Condom use at last sex” variables. In order to make these notes more obvious when first reading through the table, we now refer to them with an asterisk next to each variable. 

Reviewer #3

Main points

1. Randomisation schedule was created in an unusual way. Dyads were asked to select their randomisation treatment from a paper bag. This could have led to systematic bias and Table 1 indeed shows large imbalances for some of the participants’ characteristics at baseline (e.g. % positive for STI, mean number of STI, ethnicity, history of sex and socio-economic status). These are key common causes of both treatment allocation and incidence of STI so could have confounded the results. Authors should re-perform the analysis after controlling for all these additional imbalanced factors not just for the presence of STI at baseline. Residual confounding should be also mentioned in the limitations (lines 356 onward).

The randomization system was selected for consistency with prior research and theory [8, 9, 10]. It is unclear how participant selection of intervention arm from a paper bag could be related in a biased fashion to other baseline characteristics. All participants had an equal chance of selecting the experimental or control condition.

As stated below in response to Point 6, we agree that the results we presented about differences between arms warrant a more thorough investigation of exchangeability prior to fitting our outcome models. Instead of relying on p-values, we now examine the standardized mean differences between our treatment groups. Based on a general rule of thumb that standardized mean differences above an absolute threshold of .20 may be problematic, we considered all of our standardized differences (which were below .20) to be small, and therefore did not adjust for these variables in subsequent analyses. This is now explained on p. 12, lines 345-350.

2. Similarly, the procedure of alternating between IMARA and the health promotion program would have made the process predictable and broke down the concealment of allocation so that recruiters could have decided not to enrol, say, a problematic dyad if it was known that next allocation would have been for example control and not IMARA.

For every workshop, recruiters scheduled more than 8 dyads with the assumption that randomization would occur. Only the project director knew which arm would be delivered if fewer than 8 dyads arrived for the session. Recruiters were not informed of the decision. Hence, the process was not predictable and concealment of allocation was retained. This is clarified in the text. (see p. 6, lines 154-156).

3. Mothers and daughters in the control arm did not engage in role-plays or joint commune exercises, resulting in lack of blinding. Unclear why efforts were not made to try to implement blinding. This is an additional limitation that should be mentioned in the Discussion.

Families were told they would receive either IMARA or a health promotion program. The health promotion program was very well-received by families as indicated by their post-workshop evaluations. Moreover, we structured the health promotion program to be similar to IMARA in length and amount of time dyads spent jointly and separately. Mothers and daughters actually spent the same amount of time in joint activities, but these were focused on non-sexual health topics (e.g., developing healthy meals and exercise plans). This is now clarified in the text. That said, blinding was not possible because families knew the content of their intervention once they participated. This is now indicated in the limitation section (see pp. 19-20, lines 644-660).

4. Calculation of statistical power is cryptic. It is said that dyads were randomly assigned to the IMARA (n=118) or the health promotion control program (n=81) following an overall ratio of roughly 1.5:1 favouring IMARA to enhance analytic power to detect the treatment effect. Nevertheless, in the Methods power is given for a total sample size of N=200 participants, assuming an even split of 100 vs. 100 in the 2 groups. Authors should show the difference in power using the 1:1 vs. the 1:1.5 random allocation and explicit reasons for choosing the latter. What was the expected increase in power? In general, from the text in lines 251-258, it appears that the sample size of 199 resulted as a consequence of an inevitable attrition between baseline assessment and initiation of the intervention rather than a pre-planned design.

Thank you for highlighting these points. With regard to randomization, we now state the rationale for shifting from 1:1 to 1.5:1 explicitly in the manuscript as follows: “In general, we alternated between IMARA and the health promotion condition, but in the final year of recruitment we favored IMARA to enhance analytic power to detect treatment effects and to provide a large enough sample to evaluate mechanisms associated with key outcomes in IMARA. Although originally unplanned, these strategies led to an overall randomization ratio of approximately 1.5:1 favoring IMARA.” (p. 6, lines 156-160).

With regard to power, the original study design as registered on clinicaltrials.gov was a 1:1 randomization ratio with power calculated according to this assumption. We now explicitly state this when describing randomization on p. 12, lines 336-337. However, in response to the reviewer, we performed calculations to examine the difference in power using a 1:1 vs. 1.5:1 allocation. First we calculated power with a 1:1 allocation using an updated version of G*Power 3.1.9.7, and assuming: (a) a two-sided test, (b) alpha = .05, (c) power = 80%, (d) a relative risk of .40 for IMARA vs. FUEL, (e) STI prevalence in FUEL at follow-up of .25, and (f) equal allocation to IMARA and FUEL. Findings indicated we would need a sample size of approximately 110 participants in each arm. With a sample size of 100 participants in each group [1:1 allocation with 200 total participants], with the same characteristics, the post hoc achieved power is about 76.2%. Of note, the 80% power estimated in the funded grant (and clinicaltrials.gov) in 2009 used an older version of the statistical package. Next, we conducted the same power calculations using a 1.5:1 allocation to IMARA and FUEL, which indicated that we would need a sample size of about 128 participants in IMARA and about 85 participants in FUEL, and if we had a sample size of about 120 participants in IMARA and 80 participants in FUEL [1.5:1 allocation with 200 total participants] then the post hoc achieved power is about 77.7%. The difference between the two power calculations, 76.2% vs. 77.7%, is not substantial.

For the sake of transparency, and because analytic power should not be calculated retrospectively, we retained our original power calculation in the methods section (see p. 12, lines 337-339), but we now state directly that: “Power was not substantially altered when assessed retrospectively for a 1.5:1 allocation favoring IMARA, as was ultimately arrived at in the present study” (p. 12, lines 339-341).

5. Although similar by treatment arm, the rate of non-adherence/loss to follow-up was not negligible at 16% and 10% in the two arms. Because of that it is possible that the two groups are no longer exchangeable at 12 months. It is not an unreasonable assumption in these settings that daughters who are lost to follow-up are more likely to be have acquired STIs. Authors should perform an intention to treat analysis using the missing-failure approach. Alternatively, it is possible that the effect of the strategy could have been diluted by attrition so authors should also perform an on-treatment analysis after controlling for post-baseline confounding factors using a marginal model. This should control for all possible post-baseline prognostic factors as well as common causes of acquiring STI and risk of drop-out.

Thank you for raising these important points; we agree it was an oversight on our part not to have conducted a formal intent-to-treat analysis. We now do so in two ways, as described on pp. 12-13, lines 350-380. First, consistent with the original analytic plan registered at clinicaltrials.gov, we ran the model among only those participants who provided all STI data, and for whom STI status could thus be known at both time points. Second, we conducted a more conservative follow-up intent-to-treat analysis in which participants with missing follow-up data were coded according to their baseline STI status. This approach, also known as the last observation coded forward method, provided a more rigorous check on the original modeling. Both approaches yielded statistically significant support for IMARA. These findings are now described in the results section (pp. 16-17, lines 467-542), and acknowledged in the discussion (p. 17, line 552). 

6. Lines 264-273. The issue of baseline exchangeability between arms is ruled out in the Methods with a couple of sentences saying that there was no difference at baseline between the two groups. A p-value >0.05 for these comparisons (line 236) is meaningless and should not be reported as this p-value should be equal to 1 as the null hypothesis is true by definition in a trial. On the other hand, there are larger imbalances between baseline factors which should be described in the text and controlled for in the analysis. For example, mothers in the IMARA arm were less likely to earn <$30k per annum that mothers in the control arm and this could have explained the difference in incidence of STI at 12 months. The sentence that overall >75% (which becomes ‘almost 80%’ in line 351 of Discussion section) of mothers earned <$30k annually in both groups is inaccurate and misleading.

As stated in response to Reviewer 3, Point 1, we agree that the results we presented about differences between arms warrant a more thorough investigation of exchangeability prior to fitting our outcome models. Instead of relying on p-values, we examined the standardized mean differences between our treatment groups. Based on a general rule of thumb that standardized mean differences above an absolute threshold of .20 may be problematic, we considered all of our standardized differences (which were below .20) to be small, and therefore did not adjust for these variables in subsequent analyses. This is now explained on p. 12, lines 345-350.

7. Line 288. This same model adjusted for baseline STI status was RR=3.24, 95% [CI 1.72-6.11], p=.000. Has the direction of RR being inverted? Is this the difference of control vs. IMARA?

Thank you for the opportunity to clarify our writing. The direction of the RR was not inverted. Rather, Both RRs were from the same, single model: the first (RR = .57 with 95% CI 0.37-0.88, p=.011) referred to the effect of arm (IMARA vs. health promotion control) on the 12-month STI outcome, and the second (RR = 3.24 with 95% CI 1.78-5.93, p<.001) referred to the effect of baseline STI status on the 12-month STI outcome. We have clarified this in the manuscript. Please see also our response above to Reviewer 2, Point 8. 

8. The fact that the difference observed could be due to mediators is mentioned and several potential mediators are listed e.g. improvements in mother-daughter communication etc. Was use of condom measured in the study? Change in sexual risk behaviour seems to be the most obvious surrogate endpoint/mediator in this analysis to verify.

The observed difference could be explained by a number of potential mediators, but we are reluctant to bring these findings into the current manuscript because they are beyond to scope of the primary outcome paper, consistent with clinicaltrials.gov. We respectfully submit that a subsequent manuscript is warranted where we may thoroughly evaluate mediators of the outcome based on our theoretical framework. Indeed, we recently published a paper on the role of mental health (see Response to Reviewer 2, Point 7 above). We chose to focus this paper solely on the biological outcome rather than self-reported sexual behavior, which is likely subject to greater social desirability or response bias.

9. Line 359. American adolescent girls and their mothers and may not generalise to other groups. Patterns may differ depending on the type of caregiver. For how many dyads the caregiver was not the natural mother? Unclear why a stratified analysis by type of caregiver (mother vs. other) could not be performed.

We did not take this approach both because it was inconsistent with our original analytic plan, and because our study was not powered for such stratified analyses. We now acknowledge this as a direction for future research in the discussion section (see p. 20, lines 662-664).

10. What was the effect of SISTA SiHLE and STYLE compared to this intervention? Results of the various trials should be reported and compared in the Discussion section.

Results from SISTA are reported on p. 4, lines 92-94. Results for SiHLE are reported on p. 4, lines 94-95. Findings from Project STYLE are on p. 4, lines 99-100. We cannot compare the three interventions given significant differences in target populations, research methods, and intervention characteristics. In addition, the study’s primary outcome is not reported for Project STYLE or SISTA. That said, we have strengthened the Discussion to address these issues (see pp. 17-18, lines 559-585).

Other points

1. Lines 248-250. This stringent approach was not intended to be fully powered, but rather to provide a preliminary check of whether the pattern of results observed within the full sample replicated within the subset known to be sexually active at baseline. Indeed, it is very likely to actually be under-powered. Would have been more sensible to restrict the analysis to those who did not show STI at baseline?

In order to address this point, rather than retain the analysis on the sub-sample of youth who were STI-positive at baseline, we instead included in both of the prospective models an interaction term between arm and STI status at baseline. This allowed us to test if there was a significant moderating effect of baseline STI status on the effect of arm (IMARA vs. the health promotion control) on future STI acquisition. As described on p. 13, lines 335-379, the interaction term was non-significant in both models, failing to provide support for significant moderating effects. We thus presented findings from the models without interaction terms, although all models are available upon request. Correspondingly, we removed from Table 1 descriptive statistics on the sub-sample of girls who were STI-positive at baseline and Fig 3.

2. Line 306. Suggested rewording: This study provides preliminary efficacy data indicating that IMARA protects against incident….

The sentence has been re-worded accordingly (see p. 17, lines 544-545).

3. Line 308. The authors claim that for consistency with existing literature past STI occurrence is strongly associated with future acquisition, and so it is standard to co-vary for STI history when predicting future infection. Nevertheless, the unadjusted RR of 0.43 is mentioned here, why?

Please see our response to Reviewer 3, Point 7, above. The 43% risk (corresponding to an RR of .57) indeed represented the adjusted RR. We hope that this point is now clearer within the manuscript. 

4. Line 341. Suggested rewording: It is possible THAT girls did not understand the….

If the data are consistent with the literature, why is paragraph (in lines 341-349) needed at all? Or is it instead the case that 33% is likely to be an under-estimate of the prevalence of STI at baseline?

The wording was changed as recommended (see p. 19, line 632)

We opted to retain the paragraph because we believe it is important to (1) report when findings are replicated with previous research, (2) provide interpretations for the inconsistency, (3) underscore the value of biological outcomes relative to self-reports, and (4) underscore the need for future research to find innovative strategies to improve the validity of self-reports. However, we would be happy to remove the paragraph based on the editor’s recommendation.

 References

1. Centers for Disease Control and Prevention. HIV Among African Americans 2016 [updated February 4, 2016. Available from: http://www.cdc.gov/hiv/group/racialethnic/africanamericans/index.html.

2. Forhan SE, Gottlieb SL, Sternberg MR, Xu F, Datta SD, McQuillan GM, et al. Prevalence of sexually transmitted infections among female adolescents aged 14 to 19 in the United States. Pediatrics. 2009;124(6):1505-12.

3. Magidson JF, Blashill AJ, Blanco C. Relationship between psychiatric disorders and sexually transmitted diseases in a nationally representative sample. J Psychosom Res. 2014;76(4):322-8.

4. Wingood GM, DiClemente RJ. Application of the theory of gender and power to examine HIV-related exposures, risk factors, and effective interventions for women. Health Educ Behav. 2000;27(5):539-65.

5. Wingood GM, DiClemente RJ. The role of gender relations in HIV prevention research for women. Am J Public Health. 1995;85(4):592.

6. Wilson BDM, Miller RL. Examining Strategies for Culturally Grounded HIV Prevention: A Review. AIDS Educ Prev. 2003;15(2):184-202.

7. Brown LK, Whiteley, L., Houck, C.D., Craker, L.K., Lowery, A., Beausoleil, N., Donenberg, G. The Role of Affect Management for HIV Risk Reduction for Youth in Alternative Schools. . J Am Acad Child Adolesc Psychiatry. 2017.

8. Brown LK, Hadley W, Donenberg GR, DiClemente RJ, Lescano C, Lang DM, et al. Project STYLE: A Multisite RCT for HIV Prevention Among Youths in Mental Health Treatment. Psychiatr Serv. 2014;65(3):338-44.

9. Coleman R. Random Assignment. Designing Experiments for the Social Sciences: How to plan, create, and execute research using experiments. Thousand Oaks, CA: Sage Publishing; 2018. p. 173-210.

10. KP S. An overview of randomization techniques: An unbiased assessment of outcome in clinical research. J Hum Reprod Sci. 2011;4(1):8-11.

---

## [Editor Report · Decision Letter 1]

27 Aug 2020

PONE-D-19-32194R1

IMARA: A mother-daughter group randomized controlled trial to reduce sexually transmitted infections in Black/African-American adolescents

PLOS ONE

Dear Dr. Donenberg,

Thank you for submitting your manuscript to PLOS ONE. After careful consideration, we feel that it has merit but does not fully meet PLOS ONE’s publication criteria as it currently stands. Therefore, we invite you to submit a revised version of the manuscript that addresses the points raised during the review process.

The comments below are minor, but do merit consideration prior to publication.  

We look forward to receiving your revised manuscript.

Kind regards,

Matt A Price

Academic Editor

PLOS ONE

Additional Editor Comments (if provided):

I have a few minor suggestions to consider for this revision.

ABSTRACT: The order of two sentences was a little confusing to me, I initially read the N as the final sample size (ie, taking into consideration loss to follow up). You may wish to reword "Girls provided urine samples to test for N. gonorrhoeae, C. trachomatis, and T. vaginalis infection at baseline and 12-months. Retention at 12-months was 86% with no difference across arms. Mother-daughter dyads were randomly assigned to IMARA (n=118) or a time-matched health promotion control program (n=81)." to something like "Girls provided urine samples to test for N. gonorrhoeae, C. trachomatis, and T. vaginalis infection at baseline and 12-months. Mother-daughter dyads were randomly assigned to IMARA (n=118) or a time-matched health promotion control program (n=81). Retention at 12-months was 86% with no difference across arms."

ABSTRACT: You note that the intent to treat analysis was also significant “albeit with an attenuated magnitude”, yet in your results, the confidence intervals overlap. I suspect your study was not powered to see a difference between a 43% lower chance to contract an STI, and a 37% lower chance. I might change this to read that the ITT was "similar" to your main results, not "attenuated". Those two estimates are statistically very similar (and I would interpret them as essentially the same value).

METHODS: please do add the name of the test kits for CT/NG and TV. You don't need to add the sensitivity and specificity.

DISCUSSION: In the discussion, "STI and HIV" is often mentioned together, yet your study does not test for HIV. You do note this in your response to one of the reviewers, but I think your discussion would benefit with a short mention of this in your 'limitations' paragraph so at least the reader is aware you're thinking of this, even if you weren't able to test for it.

---

## [Author Response · Author response to Decision Letter 1]

27 Aug 2020

April 27, 2020

Matthew Price

Academic Editor

PLOS ONE

Dear Dr. Price:

Re: IMARA: A mother-daughter group randomized controlled trial to reduce sexually transmitted infections in Black/African-American adolescents (PONE-D-19-32194R1)

Thank you for the opportunity to revise and resubmit for review the above referenced manuscript. We appreciate the additional helpful comments from the reviewer. 

Below, please see the reviewer comments in bold and our responses in regular text. Changes in the manuscript are indicated by page and line numbers and they are tracked in the submitted manuscript. We sincerely hope these changes meet with your and the reviewers’ approval. 

Sincerely,

Geri R. Donenberg, Ph.D.

Professor of Medicine and Psychology ⏐ University of Illinois at Chicago

Co-Vice Chair of Research ⏐ Department of Medicine

Director ⏐ Healthy Youths Program 

Director ⏐ Center for Dissemination and Implementation Science 

Reviewer Comments

ABSTRACT

The order of two sentences was a little confusing to me. You may wish to reword "Girls provided urine samples…” to something like "Girls provided urine samples to test for N. gonorrhoeae, C. trachomatis, and T. vaginalis infection at baseline and 12-months. Mother-daughter dyads were randomly assigned to IMARA (n=118) or a time-matched health promotion control program (n=81). Retention at 12-months was 86% with no difference across arms."

The reordering of the sentence was done. Please see page 2, lines 33-35.

You note that the intent to treat analysis was also significant “albeit with an attenuated magnitude”, yet in your results, the confidence intervals overlap. I suspect your study was not powered to see a difference between a 43% lower chance to contract an STI, and a 37% lower chance. I might change this to read that the ITT was "similar" to your main results, not "attenuated". Those two estimates are statistically very similar (and I would interpret them as essentially the same value).

We changed to word “attenuated” to “similar”. Please see page 2, line 39.

METHODS

Please do add the name of the test kits for CT/NG and TV. You don't need to add the sensitivity and specificity.

We added the name of the test kits. Please see page 8, lines 163-164.

DISCUSSION

In the discussion, "STI and HIV" is often mentioned together, yet your study does not test for HIV. You do note this in your response to one of the reviewers, but I think your discussion would benefit with a short mention of this in your 'limitations' paragraph so at least the reader is aware you're thinking of this, even if you weren't able to test for it.

We added the following sentences to the limitations section of the manuscript: 

“We did not test participants for HIV, because of the low incidence among adolescent girls. However, STIs are a proxy for unprotected sexual activity, a known risk for HIV.” 

Please see page 20, lines 441-443.

---

## [Editor Report · Decision Letter 2]

11 Sep 2020

IMARA: A mother-daughter group randomized controlled trial to reduce sexually transmitted infections in Black/African-American adolescents

PONE-D-19-32194R2

Dear Dr. Donenberg,

We’re pleased to inform you that your manuscript has been judged scientifically suitable for publication and will be formally accepted for publication once it meets all outstanding technical requirements.

Kind regards,

Matt A Price

Academic Editor

PLOS ONE
---

## [Editor Report · Acceptance letter]

22 Oct 2020

PONE-D-19-32194R2 

IMARA: A mother-daughter group randomized controlled trial to reduce sexually transmitted infections in Black/African-American adolescents 

Dear Dr. Donenberg:

I'm pleased to inform you that your manuscript has been deemed suitable for publication in PLOS ONE. Congratulations! Your manuscript is now with our production department. 

Kind regards, 

on behalf of

Dr. Matt A Price 

Academic Editor

PLOS ONE